# Multi-level chirality in liquid crystals formed by achiral molecules

Mirosław Salamończyk [1,2], Nataša Vaupotič[3,4], Damian Pociecha[1], Rebecca Walker[5], John M.D. Storey[5], Corrie T. Imrie [5], Cheng Wang [2], Chenhui Zhu[2] & Ewa Gorecka[1]

Complex materials often exhibit a hierarchical structure with an intriguing mechanism responsible for the 'propagation' of order from the molecular to the nano- or micro-scale level. In particular, the chirality of biological molecules such as nucleic acids and amino acids is responsible for the helical structure of DNA and proteins, which in turn leads to the lack of mirror symmetry of macro-bio-objects. To fully understand mechanisms of cross-level order transfer there is an intensive search for simpler artificial structures exhibiting hierarchical arrangement. Here we present complex systems built of achiral molecules that show four levels of structural chirality: layer chirality, helicity of a basic repeating unit, mesoscopic helix and helical filaments. The structures are identified by a combination of hard and soft x-ray diffraction measurements, optical studies and theoretical modelling. Similarly to many biological systems, the studied materials exhibit a coupling between chirality at different levels.

[1] Faculty of Chemistry, University of Warsaw, Żwirki i Wigury 101, 02-089 Warsaw, Poland. [2] Advanced Light Source, Lawrence Berkeley National Laboratory, 1 Cyclotron Rd, Berkeley, CA 94720, USA. [3] Department of Physics, Faculty of Natural Sciences and Mathematics, University of Maribor, Koroška 160, 2000 Maribor, Slovenia. [4] Jozef Stefan Institute, Jamova 39, 1000 Ljubljana, Slovenia. [5] Department of Chemistry, King's College, University of Aberdeen, Aberdeen AB24 3UE, UK. Correspondence and requests for materials should be addressed to N.V. (email: natasa.vaupotic@um.si) or to E.G. (email: gorecka@chem.uw.edu.pl)

It is well known that in biological systems, the molecular chirality of nucleic acids and amino acids renders the formation of the helical mesostructure of DNA and proteins that finally leads to a lack of mirror symmetry of macro-bio-objects. Hierarchical structures are also inherent to simpler molecular systems like liquid crystals. The presence of a chiral center in the structure of a mesogen can dramatically alter the liquid-crystalline phase behavior and therefore, material properties. If chiral molecules form a nematic or tilted smectic phase, the neighboring molecules tend to twist with respect to each other and a helix is formed[1]. In some cases, the structure is further transformed into a more complex one, for example, into the cubic lattice of a blue phase[2] or a twist grain boundary (TGB) phase[3]. Because chiral interactions are weak, the helical pitch is long, usually hundreds of nanometers. In recent years, it became apparent that twisted structures are not unique to chiral molecules, with the discovery that although achiral, bent-core molecules or bent mesogenic dimers can form helices with a surprisingly short pitch of just a few nanometers. The twist–bend nematic ($N_{TB}$) phase[4–8] is the simplest heliconical structure, and found for a number of odd-membered dimers and for some rigid bent-core molecules[9,10]. In the $N_{TB}$ phase, molecules precess along the helix axis, on average being inclined to the axis by an angle $\theta$ (tilt angle). Whether for achiral molecules a helical structure can be multihierarchical is still unclear. Helical arrangements that are self-repeating in 3D have been proposed for cubic or tetragonal phases formed by multiple end-chain molecules[11–15]. Lamellar phases - smectics, with helical structures, were suggested[16–18] and unambiguously proved just for one material[19].

In this article, we report on two simple molecular systems in which the competing interactions lead to a complex, multi-hierarchical helical structure with complexity similar to that found in biological materials. The phase structure exhibited by the studied nonsymmetric dimers was resolved by a resonant X-ray scattering (RSoXS). In contrast to nonresonant X-ray diffraction which is sensitive only to the electron density modulations, RSoXS also detects the periodic spatial variation of the orientation of molecules[20]. In our studies, we used RSoXS at the carbon absorption K-edge; this experimental method requires a very low-energy beam (~284 eV), but is universal, as carbon atoms are present in every organic molecule. Apart from the RSoXS measurements, we also performed the nonresonant X-ray diffraction studies and investigated the optical properties of the materials. Experiments were complemented by theoretical modeling. Details of the experimental methods are given in the

Methods section and theoretical calculations are presented in the Supplementary Discussion and in Supplementary Figs. 6−9.

## Results

**Studied materials.** The molecular structures of the studied compounds, nonsymmetric dimers *D1*[21] and *D2*[19], and their phase transition temperatures are given in Fig. 1.

**Four-layer structure.** Although for a material *D1* calorimetric studies clearly revealed the enthalpy change associated with the transition between the $N_{TB}$ and smectic phase (Supplementary Fig. 1), in optical studies, this phase transition is not detectable, suggesting that in both phases, there is a similar averaging of molecular orientation in space due to the formation of the helix. In cells with a homeotropic anchoring condition, both phases are optically uniaxial. In homogeneous cells, the *N* phase has a uniform texture, while in the $N_{TB}$ phase, a stripe texture[22,23] develops and it persists into the smectic phase (Supplementary Fig. 2). However, if the sample is kept in the $N_{TB}$ or smectic phase for several minutes at a constant temperature, the stripes start to disappear and a uniform texture is formed with the light extinction direction along the rubbing direction when viewed between crossed polarizers (Supplementary Fig. 2c, d). For a sample with a uniform texture, the birefringence measurements performed on heating show that there is only a minor change of birefringence upon the transition from the smectic to the $N_{TB}$ phase (Supplementary Fig. 2e), which suggests that the conical angle (tilt) is similar in both phases. In the smectic phase, the tilt angle estimated from the birefringence measurements is ~15° (Supplementary Fig. 2e). On further heating, the birefringence increases on approaching the transition to the nematic phase, which is consistent with the decrease of the tilt angle in the heliconical phase[24].

The nonresonant X-ray measurements revealed the bilayer structure of the smectic phase; the layer spacing is close to two molecular lengths (Supplementary Fig. 3), which suggests a broken up–down symmetry of the molecular arrangement inside the smectic layers, often observed for cyano-biphenyl derivatives, due to the self-segregation effect of polar and nonpolar end groups[25]. Such a bilayer periodicity is also inherent to the local structure in the *N* and $N_{TB}$ phases, because the position of the diffuse X-ray diffraction signal in these phases coincides with the Bragg peak in the smectic phase (Supplementary Fig. 3). In the high diffraction angle range in the smectic phase, the diffuse signal, corresponding to 0.45 nm, evidences a liquid-like

D1: Iso 99.7 N 72.9 $N_{TB}$ 66.7 Sm

D2: Iso 222.3 N 172.3 SmA 99.4 $SmA_b$ 94.5 HexI

**Fig. 1** Materials studied. Molecular structures and phase transition temperatures (in deg. C, detected by calorimetric studies on cooling scan) for the compounds *D1* and *D2*

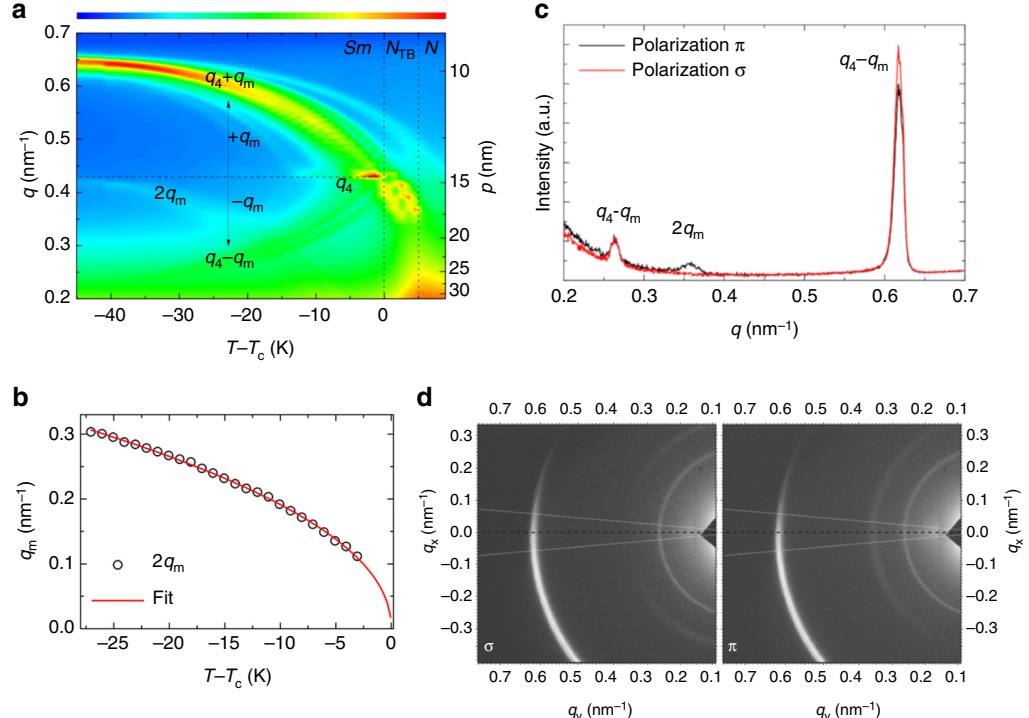

**Fig. 2** The resonant soft X-ray scattering for material D1. **a** Temperature ($T$) evolution of the scattering wave vector magnitudes ($q$) and the corresponding periodicities ($p$) of the RSoXS signals measured on heating, the intensity of the signals is coded by colors, scale bar is given above. At the transition temperature ($T_c$) from the $N_{TB}$ to the smectic phase, the signal locks at $q = q_4$, which corresponds to four smectic layer distances. This signal reflects the ideal clock-like helix in the smectic phase. For few degrees below the $N_{TB}$–smectic phase transition, the $q_4$ signal coexists with the signals $q_4 - q_m$ and $q_4 + q_m$, which are related to the distorted helical structure. The weak $2q_m$ signal is also visible. The range in which both structures, the distorted and ideal clock-like helix, coexist depends on the cooling/heating rate (see Supplementary Fig. 4). **b** The temperature dependence of the $2q_m$ signal; the red line represents the fit to the critical dependence $2q_m = 2q_{m0}T_r^{0.45}$ with $2q_{m0} = 0.6$ nm$^{-1}$, $T_r = (T_c - T)/T_c$ being the reduced temperature. **c** Intensity of signals vs. wave vector magnitude $q$ obtained by integration of the 2D RSoXS patterns (**d**) registered for the $\pi$ and $\sigma$ polarization of the incident beam. Only the areas marked by white cones in (**d**) were analyzed. The $q_4 - q_m$ and $q_4 + q_m$ signals are independent of the polarization of the incident beam, whereas the $2q_m$ signal is polarization dependent

arrangement of the molecules in the smectic layers. In the resonant X-ray scattering studies, which reveal the periodic structures related to the orientational order, the reflection associated with the helix develops at the $N$–$N_{TB}$ phase transition and moves continuously to higher values of the scattering wave vector magnitude ($q$) on reducing temperature (Fig. 2a), i.e., the pitch decreases with decreasing temperature.

At the $N_{TB}$–smectic phase transition, the RSoXS signal locks at the value $q_4$ corresponding to four molecular distances. A departure from the resonant energy even by 2–3 eV causes the signal to disappear; thus, the signal associated with the four-layer structure is purely resonant, proving its "helical" origin. In addition, the $q_2$ signal related to the bilayer periodicity is found (Supplementary Fig. 4); although this signal is resonantly enhanced, its intensity is very weak compared with the other detected signals. The continuous evolution of the structure at the $N_{TB}$–smectic phase transition suggests that the helix in the smectic phase is made of molecules rotating on the tilt cone by exactly 90° between consecutive layers. The signal due to the four-layer structure at $q_4$ persists over a few kelvin below the $N_{TB}$–smectic phase transition, and this range is determined kinetically; it is wider at lower cooling rates (Supplementary Fig. 4). On further cooling, a symmetric splitting of the $q_4$ signal into two signals: $q_4 + q_m$ and $q_4 - q_m$ is observed. The splitting increases with decreasing temperature (Fig. 2a) and both split signals have a resonant character. Their intensities are very different and for all the samples measured, the $q_4 + q_m$ signal is

the more intense. Depending on the sample, the ratio of integrated intensities of the split signals is 15–50 throughout the whole temperature range. Simultaneously with the splitting of the $q_4$ signal, an additional, very weak signal is also detected at $2q_m$ in some temperature scans (Fig. 2a). The magnitude of the scattering vector, $2q_m$ follows the critical temperature dependence (Fig. 2b), $2q_m = 2q_{m0}T_r^{\beta}$, where $T_r = (T - T_c)/T_c$ is the reduced temperature with $T_c$ being the $N_{TB}$–smectic transition temperature. The fitting parameter $2q_{m0} = 0.6$ nm$^{-1}$ is the theoretical magnitude of the modulation wave vector extrapolated to $T = 0$ K and $\beta = 0.45 \pm 0.05$ is the critical exponent associated with the phase transition. The $N_{TB}$–smectic phase transition seems to be weakly first order with the mean field critical exponent. The RSoXS signals detected in the smectic phase have different anisotropy: both split signals $q_4 + q_m$ and $q_4 - q_m$ are nearly independent of the polarization ($\sigma$ or $\pi$) of the incident X-ray beam, whereas the intensity of the $2q_m$ signal is strongly polarization sensitive (Fig. 2c, d).

## Discussion

Let us consider the possible structures responsible for the patterns observed using RSoXS in the smectic phase. For a structure with a repeating four-layer basic unit, either with a regular distribution of the molecules on a tilt cone (clock structure with 90° change of azimuthal angle between consecutive layers) or for a structure with molecules restricted to one plane (all-in-one-plane structure

with consecutive synclinic and anticlinic interlayer interfaces), only a signal at $q_4$ and its harmonics should be observed[26]. The splitting of the $q_4$ signal shows that additional modulation, with a longer periodicity, is superimposed over the basic four-layer unit. The RSoXS pattern resembles closely that for the hierarchical structure of the ferrielectric $SmC^*_{FI2}$ phase, a tilted smectic subphase detected in chiral systems in a temperature range between the synclinic and anticlinic smectic $C$ phases[27]. In the $SmC^*_{FI2}$ phase, molecules in four consecutive layers form a distorted helix, i.e., the azimuthal angle ($\delta$) between the projections of the long molecular axis to the smectic plane, in the pairs of the neighboring layers differs from $\pi/2$ (Fig. 3).

In $SmC^*_{FI2}$ phase due to the chiral interactions, the azimuthal angle in each layer increases by $\varepsilon$ with respect to the neighboring layer[28] and the whole four-layer unit cell rotates, forming a long-wavelength "optical" helix. However, it should be stressed that the materials on which we report here are achiral. Therefore, even though the structure in the smectic phase of $D1$ is similar to the $SmC^*_{FI2}$ phase, the interactions driving its formation must be different. Also, one should note that the modulation superimposed on the basic four-layer unit is much stronger than in the $SmC^*_{FI2}$ phase: deep in the smectic phase, it is of the same order of magnitude as the basic modulation, i.e., on the nanometer scale.

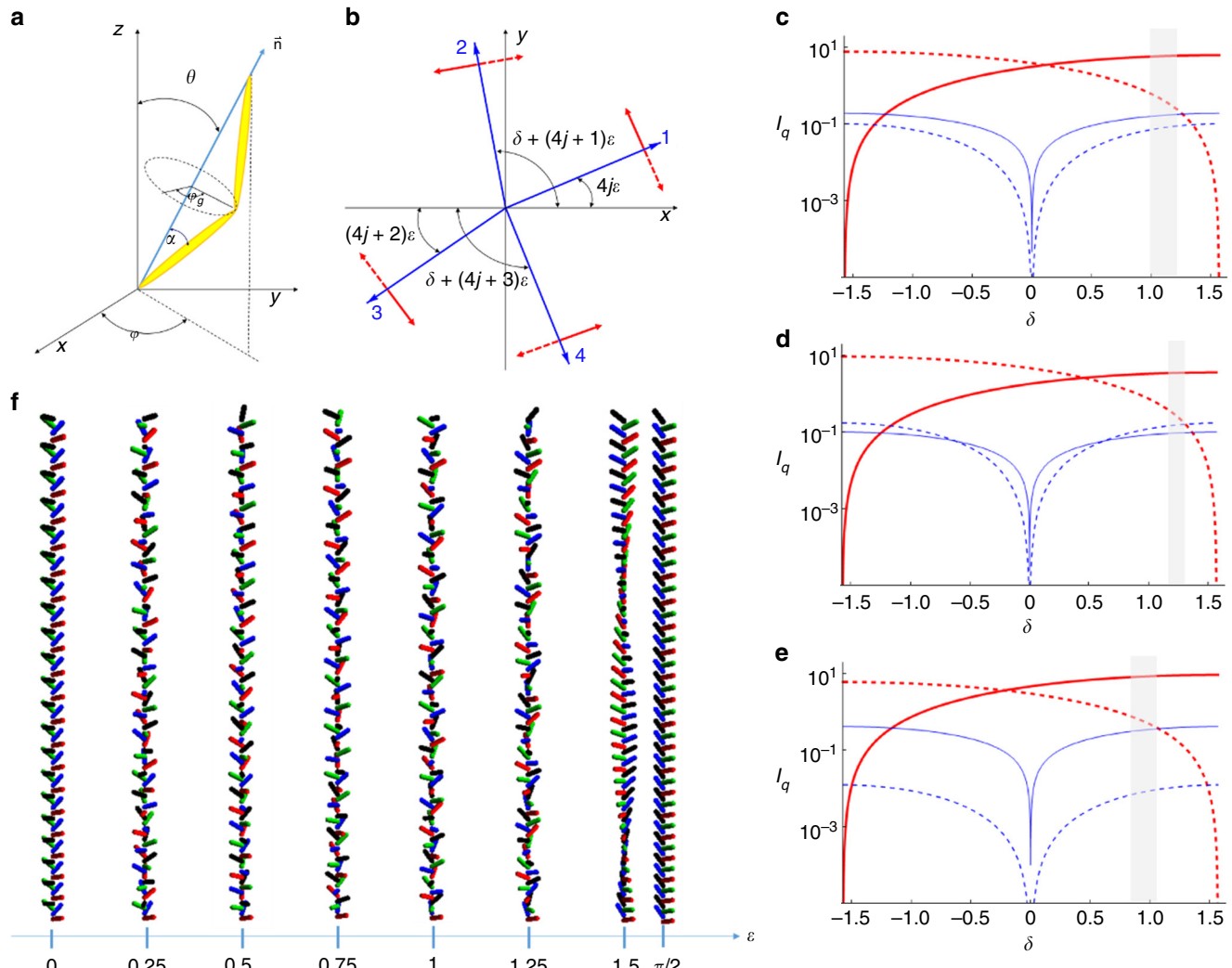

**Fig. 3** The double-helical structure in material D1. **a** The orientation of the long molecular axis (director $\vec{n}$) in the laboratory frame is described by the tilt angle $\theta$ and the azimuthal angle $\varphi$. The molecular apex angle is $\pi - 2|\alpha|$, its direction with respect to the tilt plane normal ($\vec{n} \times \hat{z}$, $\hat{z}$ being the layer normal), is marked by the angle $\varphi_g$, which is zero for a molecular tip perpendicular to the tilt plane. Two opposite layer chiralities, with a molecular apex being parallel or antiparallel to $\vec{n} \times \hat{z}$, might be expressed by the opposite signs of angle $\alpha$. **b** Projections of $\vec{n}$ on the smectic plane in the $j$th stack of four successive layers. On the "basic" four-layer structure (with $\delta$~$\pi/2$), an additional rotation by an angle $\varepsilon$ is superimposed. The signs of $\delta$ and $\varepsilon$ define the helicity of the four-layer structure and the helical structure superimposed on it, respectively. Red arrows indicate the direction of the molecular apex in the consecutive layers, solid or dashed arrows represent two possible structures with positive or negative $\alpha$. **c**–**e** The intensities ($I_q$) of the scattered light as a function of $\delta$ at $\theta = 0.25$ and $\varepsilon = 0.5$ for the peaks at $q = q_4 + q_m$ (red solid line: $\pi\sigma$- or $\sigma\pi$-polarizations), $q_4 - q_m$ (red dashed line: $\pi\sigma$- or $\sigma\pi$-polarizations), $q_2 + 2q_m$ (solid blue line: $\sigma\sigma$-polarizations), and $q_2 - 2q_m$ (dashed blue line: $\sigma\sigma$-polarizations) for **c** $\alpha = 0$ (rod-like molecule), **d** $\alpha = 0.1$, and **e** $\alpha = -0.1$. There is a contribution to the $q_2 \pm 2q_m$ peak intensities also from the $\pi\pi$-, $\pi\sigma$-, or $\sigma\pi$-polarizations, but they are orders of magnitude lower than the $\sigma\sigma$ contribution. The gray-shaded region shows the values of $\delta$ at which the ratio between the intensities of the $q_4 + q_m$ and $q_4 - q_m$ peaks is larger than 10, and at the same time, both intensities are higher than the intensities of the $q_2 \pm 2q_m$ peaks. **f** Temperature evolution of the four-layer structure of the smectic phase induced by increasing $\varepsilon$ at constant $\delta = 1.2$ and $\theta = \pi/3$. The cone angle $\theta$ is enlarged to better visualize the rotation of the molecules

To verify the proposed structure of the smectic phase, we constructed a model in which the orientation of the bent-core molecules varies from one smectic layer to another in a manner similar as in the $SmC^*_{FI2}$ phase (Fig. 3b). From the model, the RSoXS signals' intensities can be obtained by assuming that the tensor form factor describing the response of one resonant scatterer is proportional to the anisotropic part of the molecular polarizability tensor. The effect of all the carbon atoms that respond resonantly was considered by putting one resonant center in each arm of the bent molecule, with the polarizability being the largest along the molecular arm. Details of the calculation are given in the Supplementary Discussion; here we discuss the results. First, we considered the simplified case of the rod-like molecules ($\alpha = 0$): only RSoXS signals $q_4 \pm q_m$, $q_2 \pm 2q_m$, and $2q_m$ should be observed; the full pitch signal at $q_m$ is forbidden. Figure 3c gives the intensity of the peaks as a function of the distortion angle $\delta$ at fixed $\varepsilon$. The angle $\varepsilon$ that defines the additional modulation superimposed on the four-layer unit can be deduced directly from the position of the $2q_m$ peak, $\varepsilon = 2\pi(2q_m)/q_0$, where $q_0 = 2\pi/d_0$ is the wave vector associated with a single molecular layer thickness, $d_0$. Near the $N_{TB}$−smectic phase transition $\varepsilon$ is a few degrees, and it increases to nearly 45° ~40 K below the transition. The extrapolated value $2q_{m0}$ obtained from the fitting of $2q_m$ to the power law corresponds to $\varepsilon \approx 70°$. The angle $\delta$ characterizing the basic four-layer unit can be estimated from the ratio of the signal intensities; $\frac{I_{q_4+q_m}}{I_{q_4-q_m}}$ approaches 1 for an all-in-one plane model ($\delta = 0$), while it diverges for $\delta$ equal to $\frac{\pi}{2}$ (an ideal clock structure). Taking into account the experimentally detected ratio of the signal intensities that is in the range from 15 to 50 and the fact that the intensity of the $q_2 \pm 2q_m$ peaks is below the detection limit, we find $\delta \sim 60°-70°$. It is worth noting that the sign of $\varepsilon$ with respect to that of $\delta$ uniquely defines the relative intensities of the split signals. For all the measured samples, the intensity of the $q_4 + q_m$ peak is higher than that for the $q_4 - q_m$ peak, which indicates that $\varepsilon$ and $\delta$ are coupled; therefore, the handedness of the "short" four-layer helix (sign of $\delta$) determines the handedness of the "long" helical modulations (sign of $\varepsilon$). The model also correctly predicts the polarization dependence of the diffraction signals (Fig. 2c). For the $\sigma$-polarized incident light (i.e., polarization in the direction perpendicular to the scattering plane), the $q_4 \pm q_m$ peaks are $\pi$-polarized (polarization in the scattering plane) and vice versa, which is in agreement with the experimentally observed lack of split peak intensity changes upon changing the polarization of the incident beam for powder samples. On the contrary, the modulation peak, $2q_m$, is strongly polarization dependent. For the $\sigma$-incident polarization, the peak has strong $\sigma$ and weak $\pi$ components; for the $\pi$-incident polarization, there is only a weak $\sigma$-component.

Taking into account the nearly temperature-independent angle $\delta$ and a strongly temperature-dependent angle $\varepsilon$, the evolution of the helical structure can be reconstructed. The smectic phase just below the $N_{TB}$ phase has almost an ideal clock four-layer structure $(\delta \sim \frac{\pi}{2}, \varepsilon \sim 0)$. As the temperature is reduced, the value of $\varepsilon$ grows and the structure evolves toward an anticlinic one. The structures for the chosen $\varepsilon$ values and a "virtual" structure obtained by increasing $\varepsilon$ to 90° are shown in Fig. 3f and in Supplementary Movie 1.

Although the model in which molecules are represented by simple rods with the resonant atoms in the upper and lower part of the molecule appears to reproduce most of the experimental results, for the sake of completeness of a theoretical approach, we also considered a more general case. The lack of symmetry of the molecular structure was introduced by placing two "atoms" with different polarizabilities to represent the different mesogenic cores of the nonsymmetric dimeric molecule. This also yields the

resonantly enhanced $q_2$ signal related to bilayers, as observed in the experiment. The bend of the molecular structure was also considered by a rotation of polarizability tensors representing two mesogenic cores in a molecule in the opposite directions by an angle $\alpha$ (see Fig. 3a). It should be noted that the bending of molecules, which are tilted with respect to the layer normal, introduces another level of structural chirality ("layer chirality"), as changing the tip direction with respect to the tilt plane results in the mirror-reflected structure[29]. The "layer" chirality is defined in the model by the sign of $\alpha$. The additional chirality level influences the relative signal intensities (Fig. 3d, e). To account for the intensity ratio $\frac{I_{q_4+q_m}}{I_{q_4-q_m}}$ being 15:50 and for the negligible intensity of the $q_2 - 2q_m$ signal, the angle $\alpha \sim 10°$ must be taken, which corresponds to the molecular apex angle considerably larger than 120°. Most probably, also the sign of "layer" chirality ($\alpha$) is coupled to the handedness of the four-layer basic unit ($\delta$) and "long-helix" ($\varepsilon$); however, it cannot be unambiguously confirmed within the proposed model, as the proper ratio of the RSoXS signal intensities can be obtained for both positive and negative values of $\alpha$ (Fig. 3d, e).

By studying several bent dimer materials forming smectic phases, we found that a multihierarchical structure is not restricted only to compounds with the $N_{TB}$−smectic phase sequence, it was also observed for compounds of a homolog series CB6OIBeO$n$, which form a tilted hexatic I ($HexI$) phase—a phase with a short positional ordering of molecules, but a long-range correlation of the local crystallographic axis direction (bond orientational order, BOO) within the layers. The previous resonant X-ray studies for the CB6OIBeO$n$ homolog series revealed a simple "clock" helix for the twist–bend smectic phase[19], which appears above the $HexI$ phase for the shorter homologs, $n = 7$ and 8. Here we studied in detail material $D2$, a homolog with a longer terminal chain, $n = 10$, in which the hexatic phase is formed below the non-tilted biaxial orthogonal smectic phase (Sm$A_b$). In the Sm$A_b$ phase, the layer spacing is comparable with the molecular length; however, in the hexatic phase, nonresonant X-ray studies revealed a bilayer structure. Although the main diffraction signal corresponds to the molecular length, its weak subharmonic ($q_2$) related to the bilayer structure is also detected (Supplementary Fig. 5) and it shows that either a weak self-segregation effect of polar and nonpolar end groups occurs or the phase is of a general tilt type[30]. In the RSoXS measurements in the $HexI$ phase, the bilayer signal $q_2$ and split peaks $q_2 - q_m$ and $q_2 + q_m$ are seen (Fig. 4 and Supplementary Fig. 5), the signal intensity ratio (integrated) of the latter two is ~100 over the whole temperature range, with the $q_2 + q_m$ signal being the stronger one. The split signals disappear if the energy of the X-ray beam is off-resonant (Fig. 4c, d), proving that these signals are due to the modulation of molecular orientation. The splitting of the $q_2$ signal occurs simultaneously with the appearance of a low-angle $2q_m$ signal corresponding to an approximately 55-nm periodicity. The $2q_m$ signal has the same azimuthal direction as the $q_2$ signal (Fig. 4). From the splitting of the $q_2$ peak and the position of the $2q_m$ peak, we deduce that a helical modulation with $\varepsilon \sim 16°$ is superimposed over the basic bilayer structure. Interestingly, the periodicity at ~55 nm (corresponding to the $2q_m$ signal) can also be detected by AFM imaging (Fig. 5). In the low-angle range of the RSoXS pattern, a signal $q_f$ corresponding to a periodicity of 150 nm is also detected, the azimuthal position of the $q_f$ signal is orthogonal to the $2q_m$ signal. The possible origin of this signal is the pitch of the mesoscopic helical filaments formed in the $HexI$ phase (Fig. 5) - the filaments resemble those found in the $B_4$ phase[31].

To analyze the RSoXS patterns and reveal the molecular orientational structure of the $HexI$ phase of compound $D2$, we

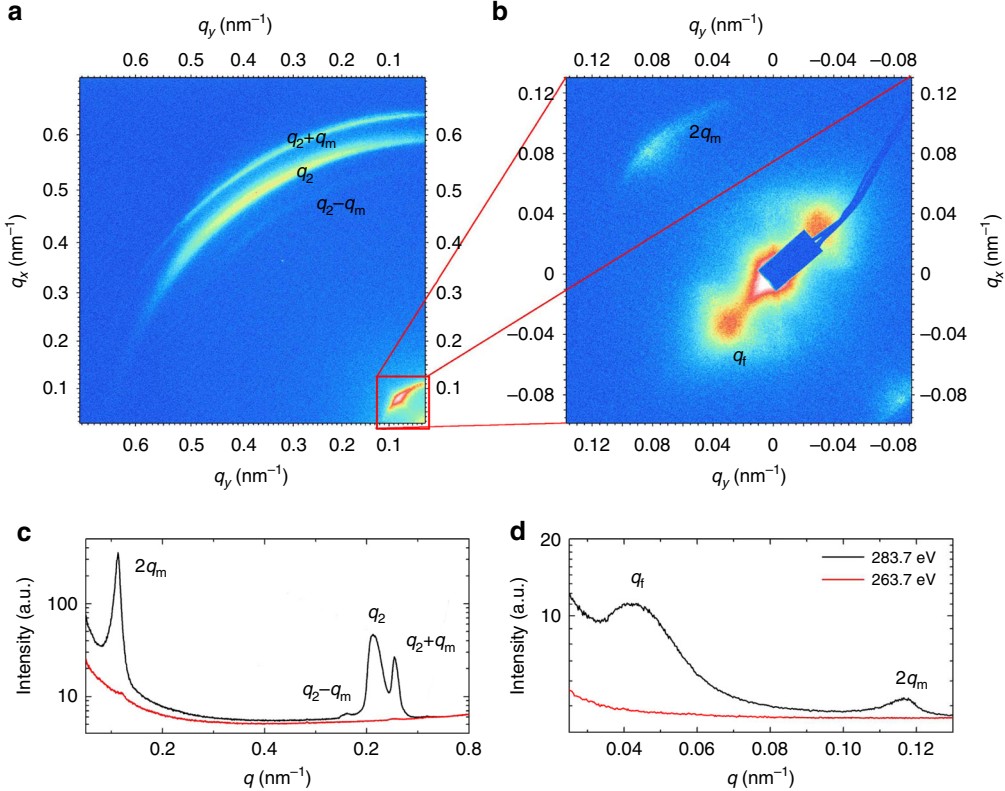

**Fig. 4** The resonant soft X-ray scattering for material *D2*. 2D RSoXS patterns in the *HexI* phase registered at 80 °C in a **a** wide and **b** small-angle regimes. The signal intensity integrated over an azimuthal angle in **c** wide and **d** small-angle range; black line: intensity registered at carbon absorption K-edge, 283.7 eV; red line: intensity registered at off-resonant energy 263.7 eV

modeled the bilayer unit in which the orientation of the molecules in the successive layers changes by an angle $\pi + \delta$, and the azimuthal position of the molecules is additionally modulated by an angle $\varepsilon$, giving a helix superimposed on the bilayer unit, as shown in Fig. 6a; details of the calculation are given in the Supplementary Discussion. For an anticlinic structure, the model predicts the absence of the full pitch, ($q_m$), peak, the peak that is not observed experimentally either; therefore, for further analysis, $\delta$ is set to zero. To account for the asymmetry in the intensity of the split signals $q_2 \pm q_m$, the molecular bend has to be considered by setting $\alpha \neq 0$, and in order to have the $q_2 + q_m$ peak more intense than the $q_2 - q_m$ peak, the $\varepsilon$ and $\alpha$ angles have to have opposite signs (Fig. 6b). Therefore, we conclude that also in the bilayer structure, the layer chirality is coupled to the handedness of the "long" helix. The experimentally determined ratio of the intensities between the $q_2 \pm q_m$ peaks that is ~100 can be obtained for $\alpha \approx -20°$ or $\alpha \approx -30°$. We have also considered a structure with the layer chirality switch between the neighboring layers ($\alpha$ of the opposite sign in consecutive layers) (Fig. 6c). In this case, the peaks $q_2 \pm q_m$ are of comparable intensities and the intensities of the $q_2 \pm 2q_m$ and $q_m$ peaks are not negligible, which disagrees with the experimental findings. The bilayer structure as predicted by the combination of the model and experimental results is shown in Fig. 5e.

In summary, both studied compounds formed smectic phases with multilevel chiral structures, despite being composed of achiral molecules. At the lowest level, "layer chirality" is defined by the direction of the molecular apex with respect to the tilt plane. Chiral layers are stacked into the two- or four-layer basic units for materials *D2* and *D1*, respectively; in the case of *D1*, such a stacking is helical. The basic structure of both

smectic phases is in addition helically modulated at a larger scale of several layers. Moreover, the material *D2* exhibits a chiral morphology—it forms mesoscopic twisted filaments. Resonant X-ray scattering studies revealed that the sign of chirality at these different levels is coupled.

Steric[32] and flexoelectric[33,34] interactions were suggested as a source of the short helix in the $N_{TB}$ phase, so we assume that these interactions are also responsible for the structure of the helical smectic phases. In the nematic phase, these interactions lead to an "ideal helix", the pitch of which decreases on reducing temperature. The formation of smectic layers leads to a distortion of this ideal clock. As the temperature is lowered and competition between the twisting due to the flexoelectricity and the entropy-driven tendency for molecules to stay in one tilt plane increases, the structure continuously evolves from nearly an ideal four-layer clock to a nearly anticlinic bilayer. For the achiral dimers studied here, this is a continuous evolution, while for the chiral systems, a set of discontinuous phase transitions between structures with 4-, 3-, and 2-layer basic units was observed[1]. For material *D2* for which interlayer interactions are strong, the system forms a double helix with a pitch nearly temperature independent in the broad temperature range.

## Methods

**Resonant x-ray experiments**. The resonant X-ray experiments were performed on the soft X-ray scattering beamline (11.0.1.2) at the Advanced Light Source of Lawrence Berkeley National Laboratory. The X-ray beam was tuned to the K-edge of carbon absorption with the energy ~280 eV. The X-ray beam with a cross-section of $300 \times 200$ μm was linearly polarized, with the polarization direction that can be continuously changed from the horizontal to vertical. Samples with thickness lower than 1 μm were placed between two 100-nm-thick $Si_3N_4$ slides. The scattering intensity was recorded using the Princeton PI-MTE CCD detector, cooled to −45 °C, having a pixel size of 27 μm, with an adjustable distance from the

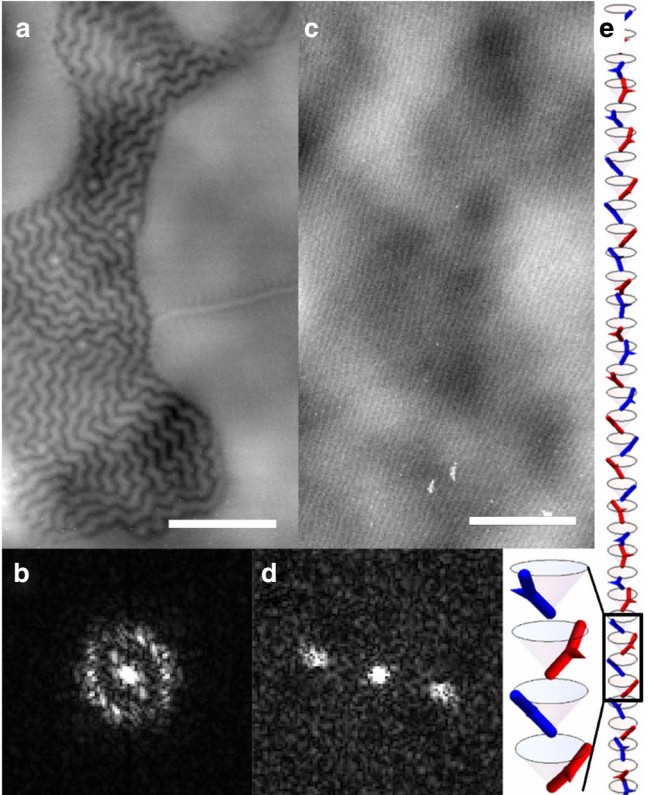

**Fig. 5** Atomic force microscopy studies of the Hexatic I phase of compound *D2* and the structure model. Two morphologies coexist in samples of the *HexI* phase of compound *D2*; the most commonly observed are mesoscopic twisted filaments (**a**); however, in samples quickly quenched to room temperature, also flat areas with parallel lines are observed (**c**). Scale bars in (**a**) and (**c**) correspond to 1 μm. FFT of the AFM images reveals the characteristic periodicities of ~150 nm and ~55 nm for filaments and lines, (**b**) and (**d**) respectively. **e** Model of the helical HexI phase; the red and blue bent rods present the molecules in the neighboring layers. The arrows are added to better visualize the direction of the molecular tip. The structure is drawn for $\theta = 0.7$, $\alpha = -0.3$, $\varepsilon = 0.3$, and $\delta = 0$. The cone angle is chosen, which is larger than that experimentally determined to better visualize the rotation of the molecules

sample. The detector was translated off-axis to enable recording of the diffracted X-ray intensity. The adjustable position of the detector allowed to cover a broad range of $q$ vectors, corresponding to periodicities from approximately 5.0 to 500 nm.

**Non-resonant X-ray**. The nonresonant X-ray diffractograms were obtained with the Bruker D8 GADDS system (CuKα line, Goebel mirror, point beam collimator, Vantec2000 area detector). Samples were prepared as droplets on a heated surface. The temperature dependence of the layer thickness was determined from the small-angle X-ray diffraction experiments performed with the Bruker D8 Discover system (CuKα line, Goebel mirror, Anton Paar DCS350 heating stage, scintillation counter) working in the reflection mode. Homeotropically aligned samples were used, prepared as a thin film on a silicon reflectionless wafer.

**Optical studies**. Optical studies were performed by using the Zeiss Imager A2m polarizing microscope equipped with Linkam heating stage. Samples were observed in glass cells with various thickness: from 1.8 to 10 μm.

**Birefringence**. The birefringence was calculated from the optical retardation at red light ($\lambda = 690$ nm). The retardation was measured with a setup based on a photoelastic modulator (PEM-90, Hinds) working at a modulation frequency $f = 50$ kHz; as a light source, a halogen lamp (Hamamatsu LC8) was used equipped with a narrow band-pass filter. The signal from a photodiode (FLC Electronics PIN-20) was deconvoluted by a lock-in amplifier (EG&G 7265) into $1f$ and $2f$ components to yield the retardation induced by the sample. Knowing the sample thickness, the retardation was recalculated into optical birefringence. For birefringence measurements, the 3-μm-thick cells were used with a planar alignment layer. Because the system allows to measure retardation only up to $\lambda/2$, the absolute value of retardation was determined by comparing the results obtained for the samples with different thicknesses, 1.8 and 5 μm and assuming a continuous evolution of retardation near the phase transition to the isotropic phase. The changes of birefringence in the nematic phase were fitted by assuming a critical dependence $\Delta n = \Delta n_0[(T - T_c)/T_c]^\beta$, where $\Delta n_0$, $T_c$, and $\beta$ were free parameters. The conical tilt angle was estimated from a decrease of birefringence in the $N_{TB}$ and smectic phase from the extrapolated value of $\Delta n$ found in the nematic phase as $\Delta n_{tilt} = 1/2 \Delta n (3 \cos^2 \theta - 1)$[24].

**AFM images**. The AFM images were taken with the Bruker Dimension Icon microscope, working in the tapping mode at the liquid crystalline−air surface. Cantilevers with a low spring constant 0.4 N/m were used, the resonant frequency was in the range of 70−80 kHz, and a typical scan frequency was 1 Hz. Samples for the AFM imaging were prepared on microscopy cover glass at an elevated temperature and quenched to room temperature.

**Calorimetric studies**. Calorimetric studies were performed with a TA DSC Q200 calorimeter, samples of mass from 1 to 3 mg were sealed in aluminum pans and kept in nitrogen atmosphere during measurement, and both heating and cooling scans were performed with a rate of 5–10 K/min.

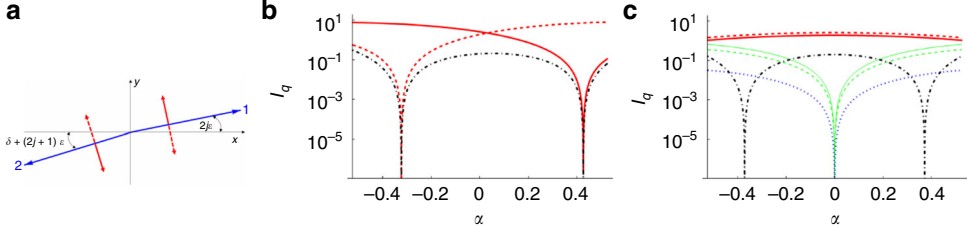

**Fig. 6** Model predictions for the bilayer structure. **a** Molecules in the successive layers differ in the direction of the long molecular axis. The figure shows the projection of the director $\vec{n}$ on the smectic plane (*xy*-plane) in the jth stack of two successive layers; on the nearly anticlinic structure of the director in the neighboring layers, the additional rotation by angle $\varepsilon$ is superimposed. Thick red (solid or dashed) arrows denote the direction of the molecular tip when the tilt is perpendicular to the tilt plane ($\phi_g$ equals 0 or $\pi$, see also Fig. 4), which defines the sign of the layer chirality. **b**, **c** The intensities ($I_q$) of the scattered light as a function of angle $\alpha$ defining the molecular bend, at $\theta = 0.4$, $\varepsilon = 0.3$, and $\delta = 0$ for the peaks at $q = q_2 + q_m$ (red solid line: $\pi\sigma$ or $\sigma\pi$-polarizations), $q_2 - q_m$ (red dashed line: $\pi\sigma$ or $\sigma\pi$-polarizations), $2q_m$ (dash-dotted black line: $\sigma\sigma$-polarizations), $q_2 + 2q_m$ (solid green line: $\sigma\sigma$-polarizations), $q_2 - 2q_m$ (dashed green line: $\sigma\sigma$-polarizations), and $q_m$ (dotted blue line: $\pi\sigma$ or $\sigma\pi$-polarizations) for the structure **b** with preserved layer chirality between the neighboring layers and **c** with the layer chirality switch between the layers. For a structure with the chirality switch, we keep on a graph both the negative and positive part of $\alpha$ to emphasize the full symmetry of the graph with respect to this parameter

## Data availability

The data that support the findings of this study are available from the corresponding authors upon reasonable request.

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

## Acknowledgements

M.S., D.P., and N.V. acknowledge the support of the National Science Centre (Poland) under the grant no. 2016/22/A/ST5/00319. E.G. acknowledges the funding from the Foundation for Polish Science through the Sabbatical Fellowships Program. N.V. acknowledges the support of the Slovenian Research Agency (ARRS), through the research core funding no. P1-0055. R.W. gratefully acknowledges the Carnegie Trust for the Universities of Scotland for funding the award of a PhD scholarship. The beamline 11.0.1.2 at the Advanced Light Source at the Lawrence Berkeley National Laboratory is supported by the director of the Office of Science, Office of Basic Energy Sciences, of the U.S. Department of Energy under Contract No. DE-AC02-05CH11231.

## Author contributions

M.S., C.Z., C.W., D.P., and E.G. performed resonant and nonresonant X-ray studies; M.S. performed AFM imaging; N.V. did theoretical modeling; D.P. and R.W. performed structural and optical studies; R.W., J.M.D.S., and C.T.I. designed and synthesized materials.

## Additional information

**Competing interests:** The authors declare no competing interests.

