## [Peer Review File · Nature Communications]

REVIEWERS' COMMENTS:

Reviewer #1 (Remarks to the Author):

The manuscript "Multi-level Chirality in Liquid Crystals Formed by Achiral Molecules" by Salamonczyk et al. describes the development of a hierarchical structure in a liquid crystalline structure formed by achiral mesogens. The work heavily relies on the resonant x-ray study at the carbon K-edge made on nonsymmetric dimers forming a twist-bend nematic phase. The authors provide a careful analysis of the scattering data and suggest a model for the mesophase structure. Although chiral symmetry breaking on different levels have already been reported in various bent-core LCs, this particular paper provides evidence for a four-layer SmC_{F12} phase and a bilayer phase. The authors demonstrate that the sign of handedness at different chirality levels are coupled.

The paper is well written, and the results are convincingly presented. In various systems, chiral order parameter couplings lead to hierarchical chiral structures. From that standpoint the findings reported here are not quite new. On the other hand, the experimental observation of the transitions between those structures in a system composed of the achiral molecules is intriguing. Some minor critics regarding the choice of expressions: line 224 "Such a nonsymmetry yields..." would perhaps be better expressed by "Such an asymmetry", line 88 "the signal... moves continuously" might be put as "the peak/reflection ... moves"

Reviewer #2 (Remarks to the Author):

The paper reports an X-ray diffraction study of two achiral molecules that form chiral liquid crystal phases. Resonance X-ray scattering, supported by modelling the structures together with optical and AFM studies are the principal methods. Chiral domains from achiral molecules are already well known. However, the novelty of this work stems from the four different levels at which chirality is observed, giving some insight into the mechanism by which molecular properties can lead to chiral materials and objects. This is an important topic and will be of interest to a wide audience. In general, the experimental method has been well described and the analysis by comparison with models is convincing. However, there is one point which does not seem to make sense. In figure 3 caption, the angle α is introduced as the tilt of one arm away from the tip-to-tip line. The apex angle is then $\pi - 2\alpha$. From this, it seems that α can only be zero or positive but later it is stated that the direction of the apex (sign of α) defines the chirality. Should this be the general tilt angle? I recommend some clarification of this point before publication.

Reviewer #1 (Remarks to the Author):

The manuscript “Multi-level Chirality in Liquid Crystals Formed by Achiral Molecules” by Salamonczyk et al. describes the development of a hierarchical structure in a liquid crystalline structure formed by achiral mesogens. The work heavily relies on the resonant x-ray study at the carbon K-edge made on nonsymmetric dimers forming a twist-bend nematic phase. The authors provide a careful analysis of the scattering data and suggest a model for the mesophase structure. Although chiral symmetry breaking on different levels have already been reported in various bent-core LCs, this particular paper provides evidence for a four-layer SmC_{FI2} phase and a bilayer phase. The authors demonstrate that the sign of handedness at different chirality levels are coupled.

The paper is well written, and the results are convincingly presented. In various systems, chiral order parameter couplings lead to hierarchical chiral structures. From that standpoint the findings reported here are not quite new. On the other hand, the experimental observation of the transitions between those structures in a system composed of the achiral molecules is intriguing.

Some minor critics regarding the choice of expressions: line 224 “Such a nonsymmetry yields...” would perhaps be better expressed by “Such an asymmetry”,

We purposely avoided the expression ‘asymmetry’, as in chemistry it is commonly used to describe molecular chirality. We have rewritten the paragraph; now it reads : “The lack of symmetry of the molecular structure was introduced by placing two ‘atoms’ with different polarizabilities to represent the different mesogenic cores of the nonsymmetric dimeric molecule. This yields also...”

line 88 “the signal... moves continuously” might be put as “the peak/reflection ... moves”

We have changed the sentence according to referee suggestion.

Reviewer #2 (Remarks to the Author):

The paper reports an X-ray diffraction study of two achiral molecules that form chiral liquid crystal phases. Resonance X-ray scattering, supported by modelling the structures together with optical and AFM studies are the principal methods. Chiral domains from achiral molecules are already well known. However, the novelty of this work stems from the four different levels at which chirality is observed, giving some insight into the mechanism by which molecular properties can lead to chiral materials and objects. This is an important topic and will be of interest to a wide audience.

In general, the experimental method has been well described and the analysis by comparison with models is convincing. However, there is one point which does not seem to make sense. In figure 3 caption, the angle alpha is introduced as the tilt of one arm away from the tip-to-tip line. The apex angle is then pi-2*alpha. From this, it seems that alpha can only be zero or positive but later it is stated that the direction of the apex (sign of alpha) defines the chirality. Should this be the general tilt angle? I recommend some clarification of this point before publication.

In the paper we have presented only the nongeneral tilt structures, with $\varphi_g=0$. In such a case a change of sign of the angle α leads to the inversion of the molecular apex direction and thus reverses the layer chirality. The value of the molecular apex angle is expressed by $\pi - 2|\alpha|$. We have rewritten the Fig.3 caption accordingly.